# ASR free End-to-End SLU using the Transformer

## Abstract

End-to-end spoken language understanding (SLU) systems directly map speech to intent through a single trainable model whereas conventional SLU systems use Automatic Speech Recognition (ASR) to convert speech to text and utilize Natural Language Understanding (NLU) to get intent. In this paper, we show how transformer-based architecture can be used for building end to end SLU systems. We conducted experiments on the Fluent Speech Commands (FSC) dataset, where intents are formed as combinations of three slots namely action, object, and location. We also demonstrate how state-of-the-art results can be obtained using a combination of various data augmentation methods.

## 1. Introduction

With the growing demand of voice interfaces for various smart devices (e.g. smartphone, smartTV, in-car navigation system) Spoken Language Understanding (SLU) has drawn a great deal of attention in recent years. Traditional SLU approaches use the text transcribed by an automatic speech recognition (ASR) system to extract the intent of the user and the slots describing the query (Mesnil et al., 2015). The main problem with Traditional SLU systems is that the errors occurred while transcribing the audio is being forwarded and affects the intent and the slot filling task. One way to avoid this problem is by combining ASR and NLU (referred as end-to-end SLU) and directly map speech to intent (Chen et al., 2018), (Lugosch et al., 2019). In this method the model is first pre-trained to predict ASR targets (words and phonemes). The word and phoneme classifiers are then discarded, and the entire model is then trained end-to-end on the supervised SLU task. The pre-trained model weights can be either frozen or fine-tuned during the SLU task training.

[1]Anonymous Institution, Anonymous City, Anonymous Region, Anonymous Country. Correspondence to: Anonymous Author <anon.email@domain.com>.

Preliminary work. Under review by the International Conference on Machine Learning (ICML). Do not distribute.

In this paper, we propose an ASR free end-to-end spoken language understanding using the transformer (Vaswani et al., 2017). The model doesnt learn any ASR level representation or use any pre-trained ASR model. We use the transformer encoder blocks with the convolution layer. Recurrent neural network (RNN) based approaches, particularly gated recurrent unit (GRU) and long short-term memory (LSTM) models, have achieved good performance for most of the tasks. But when compared with RNNs, the transformer-based encoder can capture the long term dependency better and can produce even better results. We use other data augmentations(e.g. changing pitch, reverberation, changing speed, noise injection) with SpecAugment (Park et al., 2019) (time masking and frequency masking) and get significantly low classification error compared to any other approaches. Following (Palogiannidi et al., 2019), instead of considering intents as the classes, we consider them as tuples of slots, each having an associated SoftMax layer. This technique converts a single-label classification task into a multi-label classification task and thus helps in reducing the number of classes. In the case of the Fluent Speech Command dataset, we have a three-slot tuple (action, object, location). We can say that an intent is predicted correctly if all the three slots corresponding to that intent are predicted correctly.

## 2. Related Work

(Lugosch et al., 2019) suggested a pre-training approach for end-to-end SLU models and also introduced the Fluent speech command dataset. They used a single trainable that directly maps speech to intent without explicitly producing a text transcript.They showed that by using the pre training techniques boost efficiency for both large and small SLU training sets.

(Wang et al., 2020) proposed an unsupervised pre-training approach for the SLU component of an end-to-end SLU system to preserve semantic features from large-scale raw audios. They first pretrain the AM component by using (Lugosch et al., 2019) approach and then feed the AM output to a softmax layer to get a posterior distribution. This posterior distribution is used as input of the next SLU component. (Palogiannidi et al., 2019) uses a RNN based end-to-end SLU for intent classification. Unlike (Lugosch et al., 2019),

(Palogiannidi et al., 2019) didnt make use of any ASR level prediction (e.g. phonemes,characters, words) and handle intent as tuples of slots. Additionally this approach uses various data augmentation methods and achieves state-of-the-art results. Our approach is closely related to (Palogiannidi et al., 2019), but rather than using LSTM we make use of transformer encoder blocks.

## 3. Model Architecture

The model consists of three parts: (1) Convolution layer, (2) Transformer block and (3) Classifier. The overall architecture of our end-to-end SLU model is shown in Figure 1. (Wang et al., 2019)discarded the sinusoidal positional encoding for transformers and used convolutionally learned input representations and got very decent results for the Automatic Speech Recognition task. Following these, we use a VGG-like convolution block (Simonyan & Zisserman, 2014) before the transformer encoder.The following section will describe the three parts separately.

### 3.1. Convolution layer

In order to make sense of a sequence, the model needs to know the position of each word in the sequence. To address this, the transformer uses a sinusoidal positional encoding. We replace the widely used sinusoidal positional encoding with the convolution layer. We feel that adding early convolutional layers allow the model to learn the relative positional encoding and helps the model to identify the right order of the input sequence. We used 2-D convolutional blocks with layer normalization and ReLU activation after each convolutional layer. Each convolutional block contains two convolutional layers followed by a max-pooling layer. The architecture is shown in the figure 2.

### 3.2. Transformer block

The input to the transformer encoder is the output of the convolution block. We will describe the details of the Transformer encoder block in this section.

#### 3.2.1. SCALED DOT-PRODUCT ATTENTION

Self-attention is a mechanism that relates different positions of input sequences to compute representations for the inputs. It uses three inputs namely queries(Q), keys(K), and values(V). The output of one query is calculated as a weighted sum of the values, where weights can be computed by taking the dot products of the query with all keys, divide each by $\frac{1}{\sqrt{d_k}}$, and apply a softmax function. The attention can :

$$Attention(Q, K, V) = softmax(\frac{QK^T}{\sqrt{d_k}})V$$

Where $d_k$ is the dimension of the key vector and the scalar

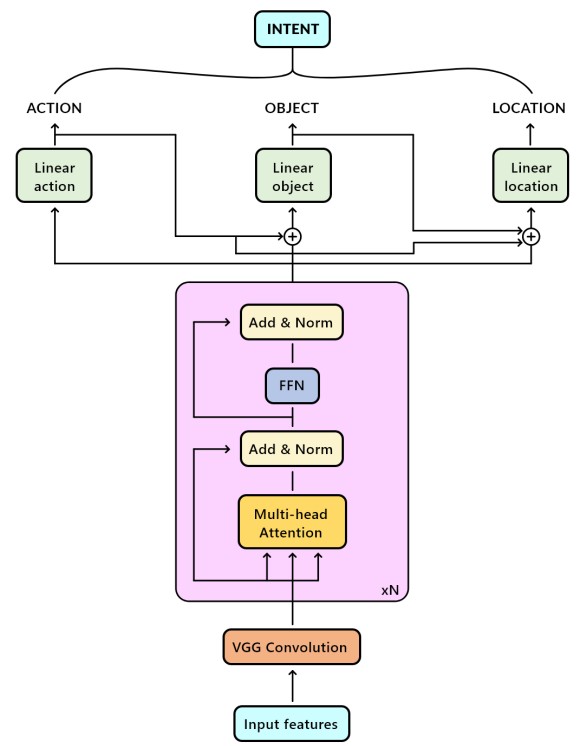

*Figure 1.* End to End SLU Architecture using Transformer

$\frac{1}{\sqrt{d_k}}$ is used to prevent softmax function into regions that have very small gradients.

#### 3.2.2. MULTI-HEAD ATTENTION

To allow the model to jointly attend to information from different representation subspaces at different positions, the transformer uses multi-head attention. Multi-head attention calculates h times scaled dot-product attention where h is the number of heads. Before performing each attention, first linearly project the queries, keys and values to more discriminated representations. Then, each Scaled Dot-Product Attention is calculated independently, and their outputs are concatenated and fed into another linear projection to obtain the final $d_{model}$ dimensional outputs. The multi-head attention can be formulated as:

$$MultiHead(Q, K, V) = Concat(head_1, ..., head_h)W^O$$
$$\text{Where } head_i = Attention(QW_i^Q, KW_i^K, VW_i^V)$$

#### 3.2.3. POSITION-WISE FEED-FORWARD NETWORK

In addition to attention, each of the encoders contains a position wise fully connected feed-forward network. It consists of two linear transformations with a ReLU activation in between.

*Figure 2.* Encoder convolution layer

$$FFN(x) = max(0, xW_1 + b_1)W^2 + b^2$$

The dimensionality of input and output is $d_{model}$, and the inner layer has dimensionality $d_{ff}$. Although the linear transformations are similar in various locations, different parameters are used from layer to layer. In addition, residual connection and layer normalization (Ba et al., 2016) are important components of the transformer. To squeeze the output of the transformer encoder, we use an average pooling layer. Besides that batch normalization (Ioffe & Szegedy, 2015) is also used.

### 3.3. Classifier

Following [8], the prediction can be made by considering both conditional and unconditional models. In case of an unconditional model, the slots are independent. The intent probability can be formulated as:

$$p(A, O, L|D) = p(A|D)p(O|D)p(L|D)$$

Here Action, Object, Location, and sequence of acoustic features for the utterance is represented by A, O, L and D respectively. In the case of conditional model, the intent probability can be formulated as :

$$p(A, O, L|D) = p(A|D)p(O|A, D)p(L|A, O, D)$$

Please note that any ordering of A,O,L is valid and there will be one independent slot and two dependent slots. When using unconditional classifiers, the slots can be predicted by using the transformer encoder output. In the case of conditional classifiers, the action slot is predicted using the transformer encoder output, whereas the object slot is predicted by considering (concatenating) both action prediction embedding and the transformer encoder output. For predicting location, we use(concatenate) action prediction embedding, object prediction embedding and the transformer encoder output. The intent predicted by the model can be then ex-

*Table 1.* Fluent Speech Commands dataset statistics

| SPLIT | SPEAKERS | UTTERANCES |
|-------|----------|------------|
| TRAIN | 77 | 23,132 |
| TEST | 10 | 3,118 |
| VALID | 10 | 3,793 |

*Table 2.* Classification error(%) on the test set, given conditional or unconditional classifier.

| CLASSIFIER | ERROR(%) |
|------------|----------|
| CONDITIONAL CLASSIFIER | 2.95 |
| UNCONDITIONAL CLASSIFIER | 3.725 |

pressed by combining the prediction for action slot, object slot and the location slot.

## 4. Experiments

In this section we are going to talk about the experiments that we conduct on Fluent Speech Command datasets. We compare our results with state-of-the-art models. We represent input signals as a sequence of 83 dimensional log-Mel filter bank features that is extracted every 10ms. We use a 512 dimensional attention vector with 4 heads along with Adam optimizer with a learning rate of 0.0001. We conducted multiple sets of experiments. Some of the experiments are conducted without using any augmentations while some use augmentation. The best epoch is chosen for each experiment based on the results on the validation set and the classification error achieved on the test set. The overall loss function for the model is the summation of cross entropy losses for the three slots.

### 4.1. Dataset

The dataset is composed of 16 kHz single-channel .wav audio files. Each audio file has a recording of a single spoken command in English. The dataset statistics are given in the Table 1. Here intents are considered as valid combinations of slots. There are 31 unique intents in total with 6,14,4 unique action,object,location respectively. For each intent there can be multiple possible wordings. For example, the intent action: "bring", object: "newspaper", location: "none" can have Bring me the newspaper, Get me the newspaper and Fetch the newspaper as the possible wordings.

### 4.2. Conditional and Unconditional classifier

To examine which classifier works best, we trained both the conditional and the unconditional model given the entire training set (without using any augmentations). Examining the results in Table 2, we observe that the model using

*Table 3.* Classification error on the test set, given conditional or unconditional classifier.

| ENCODER LAYERS | ERROR(%) |
|---|---|
| 4 | 4.45 |
| 6 | 3.49 |
| 8 | 3.07 |
| 12 | 2.95 |

conditional classifier performs better than the model using unconditional classifier.

### 4.3. Varying number of encoder layers

To explore the effect of large models, we vary the number of encoder layers. We try 4, 6, 8 and 12 encoder layers. The result of the experiments is shown in Table 3. All these experiments are conducted using the entire training set (without using any augmentations). We can see that as we are increasing the number of encoder layers, the classification error is decreasing. By using 12 encoder layers, we achieve 2.95% as the lowest classification error on the test set.

### 4.4. Data Augmentation Methods

We trained our model in three different ways. Firstly to evaluate the performance of the model on the original dataset we trained our model without using any data augmentation. We then use SpecAugment (Time masking and Frequency masking) on log-Mel filter bank features while training. To make it more robust, we first augment the original data using four different augmentations namely reverberation, pitch change, speed change, and noise injection. After using data augmentation the number of training samples increases from 23132 to 115660. We then make use of SpecAugment (Time masking and Feature masking) on log-Mel filter bank features of the augmented data while training. In this section, we are going to talk about some of the augmentation methods we used. Table 5 shows the results of augmentation.

**Noise Injection**: Noise injection is a fundamental tool for data augmentation. Adding noise during training can make the training process more robust and reduce generalization error.

**Changing Pitch**: Pitch is the quality that enables sounds to be judged as higher and lower in the sense associated with musical melodies. We use the librosa library for this data augmentation.

**Reverberation**: Reverberation is the reflection of sound waves created by the superposition of echoes. This can be done using the pysndfx library.

**Changing speed**: Changing speed is a commonly used method for doing data augmentation, where the play rate of the audio is randomly changed. Same as changing pitch, this augmentation is performed by librosa function. It stretches time series by a fixed rate. The audio speed is changed by taking a value between 0.85 to 1.15 randomly.

**SpecAugment**: (Park et al., 2019) introduced SpecAugment for data augmentation in speech recognition. SpecAugment is applied directly to the input features of a neural network. There are three basic ways to augment data which are time warping, frequency masking, and time masking. We use time masking and frequency masking methods while training the model.

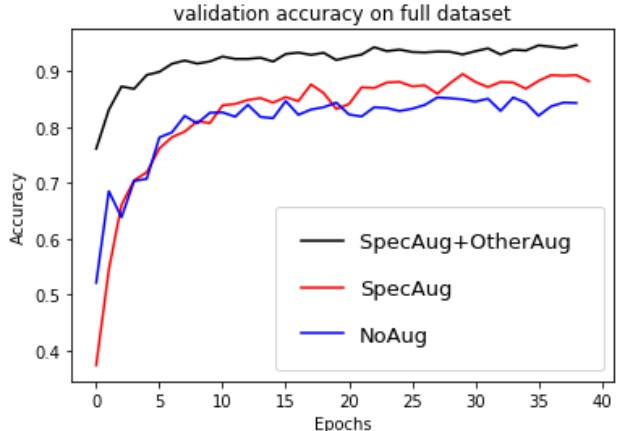

*Figure 3.* Results of training on complete dataset

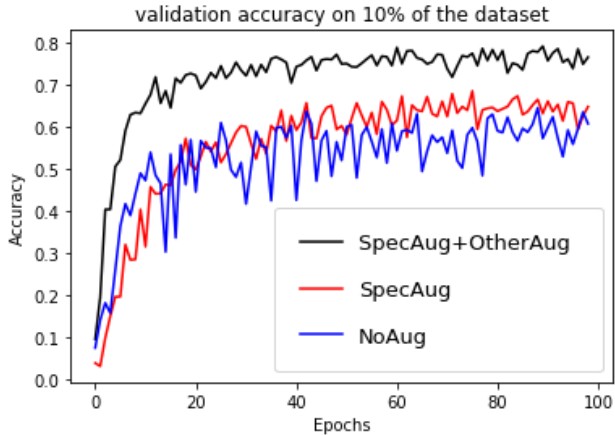

*Figure 4.* Results of training on partial dataset

### 4.5. Training on complete dataset

We conducted multiple experiments using the entire training set. Firstly we trained the model without using any augmentation. Then we experimented with SpecAugment. Finally

*Table 4.* Comparison of classification error(%) between different approaches on the Fluent Speech Command dataset.

| MODEL | ERROR(%) |
|---|---|
| PRE TRAINED SLU(LUGOSCH ET AL., 2019) | 1.2 |
| LSTM BASED SLU(PALOGIANNIDI ET AL., 2019) | 1.15 |
| ERNIE(WANG ET AL., 2020) | 0.98 |
| **SPEC AUGMENT** | **1** |
| **SPEC + OTHER AUGMENTATION** | **0.34** |

*Table 5.* Classification error(%) on full training set and 10% of the training set.

| EXPERIMENT | FULL DATA | 10% DATA |
|---|---|---|
| NO AUG | 2.95 | 25.05 |
| SPECAUG | 1 | 14.12 |
| SPECAUG + OTHERAUG | 0.34 | 8 |

we used other data augmentation methods (described earlier) with the SpecAugment and achieved a classification error of 0.34% on the test set. In comparison with the previous state-of-the-art results Table 4, our model achieved significantly low classification error. We performed all these experiments using 12 encoder layers. The validation accuracy for these experiments over time is shown in Figure 3. The results obtained on the test set for different experiments is shown in the Table 5 (Full training set column).

### 4.6. Training on 10% dataset

To evaluate the performance of models, we randomly selected 10% of the training data and used this dataset for training instead of using the full dataset. All the experiments. We conducted multiple experiments using 10% of the training set (all the experiments described for the full dataset), and observed that by using other data augmentation methods with the SpecAugment we achieved a classification error of 8%. The validation accuracy for these experiments over time is shown in Figure 4. Table 5 compares the results obtained on a full training set with the results obtained using only 10% of the training data.

## 5. Conclusion

End-to-end SLU approaches provide a new perspective for various applications since the speech is directly map to intent. In this paper, we proposed an end-to-end transformer based SLU for intent classification. The experiment results show that our proposed approach significantly outperforms SOTA end-to-end SLU systems. In the future, we plan to explore the limitations of end-to-end SLU and will try to enhance the architecture.

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
