# OpenReview forum: "ASR Free End-to-End Spoken Language Understanding using Transformers"
_ICML.cc/2020/Workshop/SAS — Submitted to SAS 2020_

### Official Review · AnonReviewer2 · 2020-06-23
**Good results reported on end-to-end SLU but the paper should be rewritten in better English and clarify several points**

**Rating:** 4
**Confidence:** 4

**Review:**

The authors propose the use of transformer instead of RNN for end-to-end SLU. After data augmentation state-of-the-art results are achieved. Good results are reported, but the contribution in not very clear. It seems to be based on top of another paper (from which the slot tuples idea and the data augmentation methods are borrowed) and contribute in changing the DNN architecture. There is thus some limited originality. The authors should clarify more in details their contributions on top of the (Palogiannidi  et al., 2019) paper which is their reference, e.g. are all the data augmentation techniques borrowed from the former paper?

The major issue with this paper though is the clarity of  presentation and the use of English. The paper should be revised and English writing should be improved. Many typos - a few are corrected in the current review, but I stopped correcting them here because of their high volume - ,awkward phrasing and transitions between sentences, grammatical errors, no gaps after full stops and parentheses,…

Some detailed comments per sections:
Section 1:
Typos - The errors…is being  forwarded and affects ->the errors are being forwarded and affect
doesnt ->doesn’t
the long term dependency -> the long term dependencies
significantly low classification error compared to…-> significantly lower classification error compared to…

“instead of considering intents as the classes, we consider them as tuples of slots, each having an associated SoftMax layer. This technique converts a single-label classification task into a multi-label classification task and thus helps in reducing the number of classes. “ -> You mean each slot having its own softmax layer, thus 3 softmax layers, right? Please clarify, not clear as is. Maybe also give some numbers, how many intents vs how many classes in each slot of the tuple. This info is given later in the paper, but would be better to clarify earlier on.

“compared to any other approaches. “ ->Very vague. Please be specific. Did you compare with certain state-of-the-art approaches and got better results?

Section 2:
“a single trainable “->what is a single trainable? Maybe there is a word missing?
Many typos and grammatical errors in paragraph 1. Please correct.
“Additionally this approach uses various data augmentation methods and achieves state-of-the-art results. Our approach is closely related to (Palogiannidi  et al., 2019), but rather than using LSTM we make use of  transformer encoder blocks. “ ->Are you using the same data augmentation methods ad (Palogiannidi et al, 2019)? Please clarify this here and in the introduction to make your contribution clear.

Section 3:
“got very decent results “ -> Vague. Did the results were as good/better/worse than using the positional encoding?

Typos in Fig.2 -> Please correct.
Explain FFN in Figure 2. Is it the same as in Figure 1? In the text it is described differently in my understanding.

You use both convolution and convolutional block/layer/.. in the text. Choose one term.

3.2.1 Rewrite, hard to follow in spite of being a well-known technique.

3.3 “Following [8], “ ->What is [8]? Citations use a different format in the rest of the text

Is the second equation of 3.3 like a hierarchical model? How do you choose which slot depends on which other/others?

Section 4:
Are the improvements statistically significant? Please report statistically significance tests.

Which of the results in Table 3 are with conditional  and which with unconditional classifier?

Section 4.6 “All the experiments. “ Not a sentence, please correct

---

### Official Review · AnonReviewer3 · 2020-06-25
**A transformer-based end-to-end architecture for spoken language understanding (SLU)**

**Rating:** 4
**Confidence:** 5

**Review:**

A transformer-based end-to-end architecture for spoken language understanding (SLU) is proposed for detecting
intents composed of three concepts: action, object, and location in the Fluent Speech Command dataset.


quality

The task has a limited semantic content mentioned in a complex acoustic environment.
Concept mentions are simpler than the ones used in many published SLU papers.

originality

The proposed architecture has transformer encoder blocks with the convolution layer.
The motivation for this choice is that transforms capture long term dependency better than recurrent neural network (RNN).
The same motivation has been reported for published experimental results in automatic speech recognition (ASR), language modeling (LM) and other natural language processing (NLP) tasks.
Transformers have been recently used for SLU.

significance

Other contributions of this paper are data augmentations consisting in changing pitch, reverberation,
changing speed, noise injection.

The related work section ignores relevant published papers on end to end SLU, and intent detection.

The architecture has three components, namely convolution layer,
transformer block and classifier.
There is nothing new in the formulation of the three components.

The paper is clearly written with an exception.

Experimental results show interesting advantages obtained by introducing augmentations.

The explanation in Table 4 is not clear.

“In comparison with the previous state of- the-art results Table 4, our model achieved significantly
low classification error”

It is not clear if the results of the competitors have been obtained with the same augmentation data.

pros
the use of augmentations

cons
little novelty

---

### Official Review · AnonReviewer1 · 2020-06-30
**An end-to-end approach for SLU using Transformers**

**Confidence:** 5
**Rating:** 6

**Review:**

This paper presents an end-to-end approach for Spoken language understanding using Transformer models. The work extends the approach of (Palogiannidi et al., 2019) who employ a similar approach to SLU using LSTMs. The key findings are that transformers can yield slightly better results than that reported using LSTMs (Palogiannidi et al., 2019). Additionally, the paper shows that specaug and other data augmentation techniques help improve performance. Overall, the approach is reasonable.

The paper does answer some questions satisfactorily:
- How does convolution compare with positional embeddings empirically?

- In Table 2, does the order of Action/Object/Location matter for the conditional model ? Which of these orders gives the best results? It would be useful to perform an ablation study.

- Since Palogiannidi et al (2019) do not employ spectral augmentation, I wonder how the results in Table 4 would change if both techniques were to use exact same type of augmentation.

---

### Decision · Program_Chairs · 2020-07-01

**Decision:**

Reject

**Comment:**

Dear author(s),

Thank you very much for your submission at the ICML2020@SaS workshop (https://icml-sas.gitlab.io/). Based on the scores assigned by the reviewers, we regret to inform you that the paper was rejected. We got 26 submissions and we were only able to accept 13 papers. We invite you anyway to consider the feedback of the reviewers and to follow our upcoming workshop on July 17.